# Severe Juvenile-Onset Systemic Lupus Erythematosus: A Case Series-Based Review and Update

**DOI:** 10.3390/children10050852

**Published:** 2023-05-10

**Authors:** Sergi Huerta-Calpe, Ignacio Del Castillo-Velilla, Aida Felipe-Villalobos, Iolanda Jordan, Lluïsa Hernández-Platero

**Affiliations:** Pediatric Intensive Care Unit, Hospital Sant Joan de Déu, 08950 Barcelona, Spain; sergi.huerta@sjd.es (S.H.-C.); ignacio.delcastillo@sjd.es (I.D.C.-V.); aida.felipe@sjd.es (A.F.-V.); lluisa.hernandez@sjd.es (L.H.-P.)

**Keywords:** juvenile-onset systemic lupus erythematosus, diffuse alveolar hemorrhage, cerebral vasculitis, antiphospholipid syndrome

## Abstract

Juvenile-onset systemic lupus erythematosus (jSLE) is a multisystemic disease diagnosed in young patients based on the clinical criteria of the European League Against Rheumatism (EULAR) and the American College of Rheumatology (ACR). The importance of this condition lies in its greater aggressiveness compared with lupus diagnosed during adulthood (aSLE). Management, which is based on supportive care and immunosuppressive drugs, aims to reduce the overall disease activity and to prevent exacerbation. Sometimes the onset is accompanied by life-threatening clinical conditions. In this paper, we introduce three recent cases of jSLE that required admission to the Pediatric Intensive Care Unit (PICU) of a Spanish pediatric hospital. This manuscript aims to review some of the main complications associated with jSLE, such as diffuse alveolar hemorrhage, cerebral vasculitis, or an antiphospholipid syndrome; these are life-threatening conditions but they have a chance of favorable prognosis if treated early and aggressively.

## 1. Introduction

Juvenile-onset systemic lupus erythematosus (jSLE) is a severe, chronic autoimmune disease with multi-system impairment that is diagnosed in people below the age of 18. This condition has an incidence of 0.3–0.9 per 100.00 children years [1] and represents approximately 15–20% of SLE patients [2], with a peak age of onset at 12.6 years [3]. In the pathophysiology of jSLE, the patient’s genetic background plays a central role. Indeed, it has been suggested that the genetic contribution has greater weight in pediatric patients than in adults. Over the last decade, some genomic regions related with different cellular immune processes (B and T cell activation, inflammation, and immune complex processing or neutrophil and monocyte signaling) have been related to an increased susceptibility to jSLE [4]. However, in most cases, additional factors (e.g., environmental, immunoregulatory, hormonal, and drug-related factors) contribute to the final development of the disease by activating a dysregulated immune response, resulting in tissue damage. The low prevalence of jSLE leads to a paucity of evidence-based data and, consequently, a lack of clinical guidelines for specific management. Currently, there are no specific diagnostic criteria for jSLE. Thus, the diagnosis is based on the adult-onset systemic lupus erythematosus (aSLE) classification criteria (Appendix A), which were developed in 2019 by the European League Against Rheumatism (EULAR) and the American College of Rheumatology (ACR) [5]. The diagnosis can be challenging, given the clinical heterogeneity of patients.

To homogenize the clinical management of these patients, in 2017, the SHARE initiative was developed by a large panel of European rheumatologists. It represents the latest European evidence-based recommendations for management of jSLE [1]. Although lupus is more prevalent in females, this predominance is less pronounced in the pediatric population. jSLE carries greater morbidity and mortality than aSLE. Specifically, a higher incidence of renal, cardiovascular, and neurological impairment at the time of diagnosis has been described [5,6]. A common characteristic feature of jSLE and aSLE is the detection of certain autoantibodies, mainly antinuclear antibodies (ANA). Anti-double-stranded DNA (anti-dsDNA), anti-Sm, anti-RNP, anti-Ro/SS-A, or anti-La/SS-B are also commonly detected [1]. Notably, within the jSLE population, children diagnosed at younger ages have more atypical presentations and worse prognoses, although their immunological profiles tend to confer lower clinical criteria scores [7]. Treatment assists with the remission or reduction of the disease activity, as well as with the prevention of exacerbation and long-term damage. Management is based on supportive treatment and drugs that modulate the immune response, such as glucocorticoids. The onset of jSLE may be severe and can require admission to a hospital’s Pediatric Intensive Care Unit (PICU), due to a range of life-threatening clinical features.

This review aims to define the baseline characteristics and clinical management of three patients who were admitted to the PICU of a tertiary pediatric hospital as a result of a life-threatening jSLE onset. The study and definition of the clinical profile and the evolution of these patients may help with the treatment of future patients since, as already mentioned, there are few clinical recommendations and guidelines regarding the management of pediatric patients.

## 2. Case Presentations

### 2.1. Case 1

A 16-year-old female was admitted to the PICU with fever, shortness of breath, and massive hemoptysis. A chest radiography showed bilateral cottony infiltrates, suggestive of diffuse alveolar hemorrhage (DAH; Figure 1). Despite an early initiation of mechanical ventilation, she had a poor outcome, with the development of severe hypoxemic respiratory failure that required high-frequency ventilation. At this point, empirical antibiotic treatment was initiated, due to the possibility of an infectious etiology. At the same time, the patient gradually developed arterial hypertension and oliguric renal failure, requiring continuous renal replacement therapy. Additionally, she had analytical alterations, such as leukopenia and thrombocytopenia, hypocomplementemia (decreased levels of C3 and C4), and positivity of some biomarkers [ANA > 1:640, anti-B2-glycoprotein (anti-B2GP1) and anti-dsDNA]. A renal biopsy showed signs of grade IV lupus nephritis and, in conjunction with clinical and immunological parameters, a diagnosis of jSLE was made (EULAR score 28). A bronchoscopy was performed on the fifth day of admission, which did not reveal any active bleeding or underlying infection. The anatomopathological examination revealed erythrophagocytosis and neutrophilia. Once the infectious etiology was ruled out, she was given aggressive treatment with high doses of methylprednisolone (1 g/kg per five days), hydroxychloroquine, cyclophosphamide, and plasmapheresis (one session every 24 h for the first 3 days; thereafter, one session every 48 h until seven sessions were completed). As there was no improvement after seven days of admission, it was decided to administer three new pulses of methylprednisolone, one dose of immunoglobulin, and one dose of rituximab. The patient also received empirical antibiotic therapy until the infection was ruled out. Finally, she exhibited good responsiveness that allowed the progressive withdrawal of respiratory and renal supports. She is currently fully recovered, with no pulmonary or renal sequelae, and she maintains immunosuppressive treatment with hydroxychloroquine, mycophenolate, and prednisone.

### 2.2. Case 2

A 16-year-old female was admitted to the PICU due to shortness of breath accompanied by fever, arthralgias, high blood pressure, and recent-onset oral ulcers. She had a family history of maternal death due to complications associated with aSLE. Chest X-ray (Figure 2) and cardiopulmonary ultrasound revealed the presence of pleural and pericardial effusion, requiring the placement of a pericardial drain due to progressive hemodynamic compromise. She presented analytical alterations, such as thrombocytopenia, hypocomplementemia (decreased levels of C3 and C4), ANA > 1:640, and positivity of highly specific antibodies such as anti-dsDNA and anti-Sm antibodies. Considering all clinical and analytical parameters, the final diagnosis of jSLE was made (EULAR score 25). An empirical antibiotic regimen was initially started, but it was withdrawn when cultures were negative. In addition, a bronchoscopy was performed and did not reveal bleeding or signs of infection. Once an active infection was excluded, she started treatment with high doses of methylprednisolone (1 g per day for five days), hydroxychloroquine (200 mg per day), and mycophenolate (720 mg/12 h). She exhibited a good initial response, with the disappearance of both effusions and the resolution of shortness of breath. However, on the tenth day of admission, she exhibited progressive loss of sensitivity, with hyporeflexia in the lower extremities, which correlated with signs of motor-sensory axonal polyneuropathy in the electromyogram. Given the detection of peripheral neuropathy, the study was extended with a brain MRI that diagnosed cerebral vasculitis associated with jSLE (Figure 3a,b), and the patient was newly treated with a pulse of methylprednisolone (500 mg) followed by tapering doses of steroids, cyclophosphamide (500 mg/m^2^, and immunoglobulins (400 mg/kg per five days). She showed a good response, with the disappearance of all neurological symptoms. She is currently asymptomatic and undergoing immunosuppressive treatment with hydroxychloroquine, prednisone, and cyclophosphamide. Additionally, she receives acetylsalicylic acid as an antiaggregant preventive therapy, given her background of vasculitis.

### 2.3. Case 3

A 12-year-old female was admitted to the PICU due to several focal seizures. A frontal venous infarction associated with thrombosis of the superior longitudinal venous sinus was diagnosed by CT (Figure 4a). Blood tests revealed signs of hemolytic anemia, as well as hypocomplementemia (decreased levels of C3 and C4), ANA > 1:640, and positivity of specific biomarkers (anti-B2GP1 > 40, lupus anticoagulant, anti-dsDNA, and anti-Sm antibodies). Given the thrombotic background and in conjunction with the immunological profile and analytical findings, a diagnosis of antiphospholipid syndrome (APS) associated with jSLE was made (EULAR score 21).

The initial treatment was a pulse of methylprednisolone (1g per day for five days), hydroxychloroquine (200 mg/24 h), plasmapheresis, immunoglobulins (400 mg/kg), cyclophosphamide (500 mg monthly), rituximab (375 mg/m2 every 15 days), and anticoagulation with low-molecular-weight heparin. After a few hours, the patient presented with refractory intracranial hypertension, requiring a percutaneous thrombectomy of the longitudinal sinus. The procedure resulted in significant improvement, allowing the withdrawal of aggressive measures. Despite this improvement, on the fifth day, she presented a new worsening due to an increase in frontal ischemic lesions (Figure 4b), requiring medical therapeutic measures, placement of a ventricular drain, and the performance of a decompressive craniotomy.

After that, she presented a good clinical evolution, eventually leading to the definitive withdrawal of all the therapeutic measures. Currently, the patient has no neurological sequelae, except for minimal left hemiparesis. She maintains immunosuppressive treatment with prednisone and hydroxychloroquine, as well as antiaggregation with acetylsalicylic acid.

Figure 5 shows the treatments received by the three patients throughout their admissions until recovery.

## 3. Discussion

The diagnosis and treatment of SLE is a challenging task for a physician. jSLE shares its pathogenesis with aSLE, although its presentation is usually more aggressive. In this sense, its association with nephritis, thrombocytopenia, and positive anti-dsDNA antibodies suggests higher disease activity and a worse overall prognosis [8].

In our institution, physicians use the current EULAR clinical criteria for diagnosis. All three of our patients were female teenagers of Latin American ethnicity, one of whom had antecedents of SLE in the family. In addition, each of them had a high immunological activity that resulted in a high EULAR score (>21 for each of them). The main life-threatening conditions were a case of DAH, a pleural effusion associated with cerebral vasculitis, and a case of stroke in the context of APS. Only one of the patients had kidney involvement, requiring renal purification. Each of the patients required aggressive medical treatment for adequate control of disease activity.

DAH is an uncommon but devastating complication of jSLE, with an estimated mortality of 20%. It is characterized by interalveolar bleeding from damaged pulmonary vasculature and clinically manifested by dyspnea, anemization, changing radiographic infiltrates, and, in half of the cases, hemoptysis [9]. If this disorder is suspected, it is mandatory to rule out any associated infection, as well as other conditions such as coagulopathy, mitral stenosis, or drug exposure. Concerning autoimmune entities, DAH can be a manifestation of pathologies that differ from jSLE, such as granulomatosis with polyangiitis, microscopic polyangiitis, Goodpasture syndrome, APS, or idiopathic pulmonary hemosiderosis. In their study, Kasap-Demir et al. compiled the casuistry of DAH associated with SLE, based on the published literature, highlighting the heterogeneity in terms of the therapeutic management received and the clinical evolution [10]. Although the optimal management of DAH has not yet been established, mechanical ventilation is necessary in most cases. Likewise, the literature published to date suggests the use of methylprednisolone pulses as soon as possible, reserving other treatments, such as cyclophosphamide, mycophenolate, plasmapheresis, or rituximab, for refractory cases [11]. Additionally, pulmonary administration of human-recombinant activated factor VIIa and tranexamic acid might be effective [12].

Vasculitis is a well-known feature of jSLE, which is defined by the presence of inflammatory changes in vessels of any size and location. Symptomatology arising from cerebral vasculitis may be nonspecific and might overlap with other characteristics of the disease. If neuropsychiatric pathology is present in a patient with jSLE, a possible contribution of APS versus vasculitis should be considered. While APS is associated with high levels of antiphospholipid antibodies, SLE flares with vasculitis are more frequently associated with hypocomplementemia, leukopenia, and anti-ds-DNA antibody positivity. Brain MRI represents the cornerstone for its confirmation and excludes other entities, such as infections, drug-induced vasculitis, malignancy-associated vasculitis, non-vasculitic inflammatory brain diseases, demyelinating disorders, and antibody-mediated inflammatory brain diseases. According to the SHARE initiative for the diagnosis and treatment of jSLE [1], the diagnostic approach should include cerebrospinal fluid examination, electroencephalogram, neuropsychological assessment of cognitive function, ophthalmologist review, nerve conductional studies, and MRI scanning, although the latter may not be fully diagnostic. In addition, cerebral angiography may provide false negatives if small vessels are involved and, in some cases, a brain biopsy is required. In addition to cerebral involvement, there may be peripheral neuropathies associated with jSLE. These are usually symmetrical polyneuropathies with sensorimotor impairment. Treatment of peripheral nervous system vasculitis usually includes high doses of corticosteroids and cyclophosphamide, although these recommendations are based on aSLE case series publications.

APS itself is an autoimmune disorder, although it is very frequently associated with other autoimmune pathologies such as SLE. It is characterized by the rapid development of multiple arterial and venous thrombosis in patients with antiphospholipid antibodies as lupus anticoagulant, anti-B2GP1, or anti-cardiolipin antibodies. Thrombosis of the lower extremities, brain, or lung are especially frequent in children. In addition to the thrombotic symptomatology, this disorder can involve non-thrombotic findings such as hemolytic anemia, thrombocytopenia, nephropathy, valvular heart disease, or transverse myelitis. Its management is based on the control of risk factors and the use of anticoagulant drugs, in addition to the immunosuppressants if it is associated with SLE. Although it is uncommon in children, there is a potential risk of suffering a catastrophic antiphospholipid syndrome (CAPS), a rapidly progressive form of APS that involves the development of microvascular thrombosis in three or more different organs and systems (ischemic stroke, pulmonary embolism, myocardial infarction, renal ischemia, etc.). This condition affects a small minority of all patients with APS and has an estimated mortality of 30% [13]. Although all organs and systems can be affected, SLE-associated CAPS entails a higher risk of cardiac or cerebral involvement. Treatment should be aggressive. Although the published literature about this subject is minimal in regard pediatric cases, the published cases suggest that the best treatment regimen is anticoagulation and high-dose corticosteroids. Additionally, plasmapheresis, immunoglobulin, and rituximab have been proposed as effective therapeutic options.

Diagnosis of jSLE is challenging, due to the possible multisystemic involvement. Although the onset can be severe, early and aggressive treatment can be successful. The treatments currently proposed were effective in our three patients in terms of survival and recovery without major sequelae. Nevertheless, well-defined algorithms are still lacking, and further powered randomized clinical studies focused on the pediatric population are needed to define the best therapeutic strategies for patients with life-threatening complications related to jSLE.

## Figures and Tables

**Figure 1 children-10-00852-f001:**
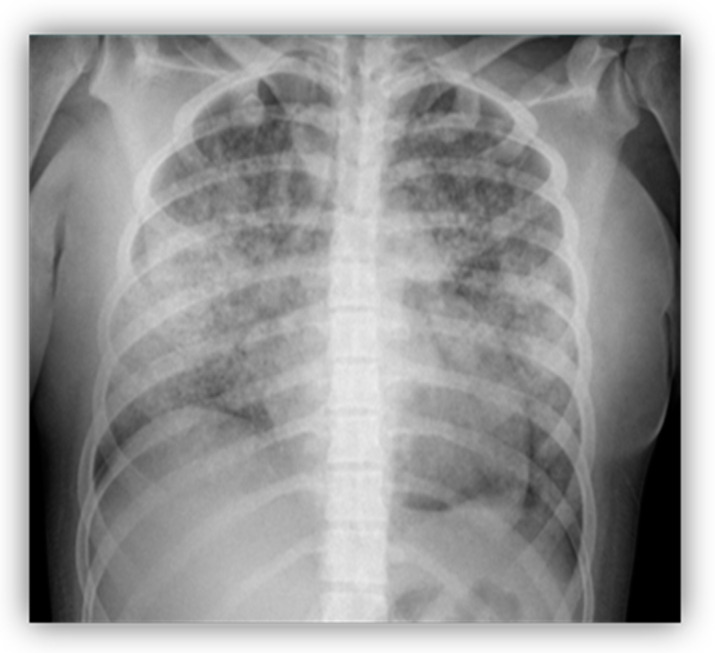
Bilateral cottony infiltrates suggestive of diffuse alveolar hemorrhage (DAH).

**Figure 2 children-10-00852-f002:**
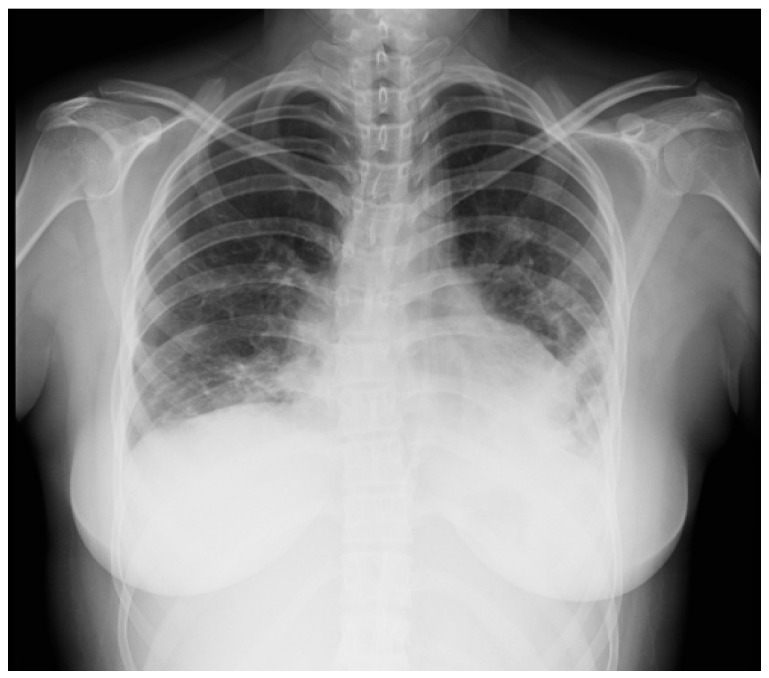
Chest X-ray suggestive of bilateral pleural effusion.

**Figure 3 children-10-00852-f003:**
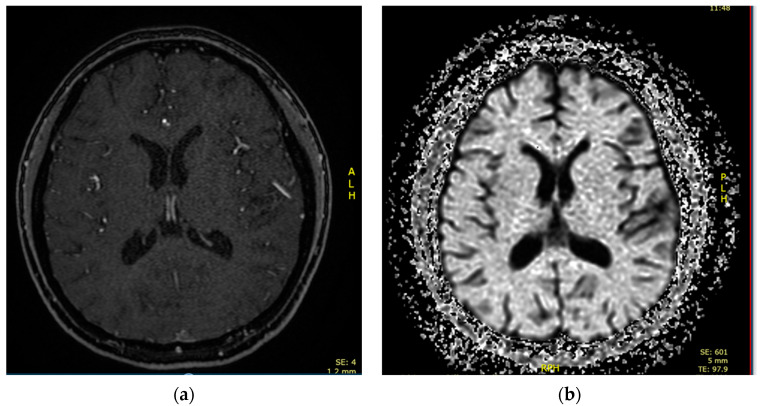
(**a**). Brain MRI showing multiple ischemic microlesion foci in fronto-parietal cortical regions and white matter of semiovale centers. (**b**) Diffusion sequence showing areas of diffusion restriction in fronto-temporal cortical areas.

**Figure 4 children-10-00852-f004:**
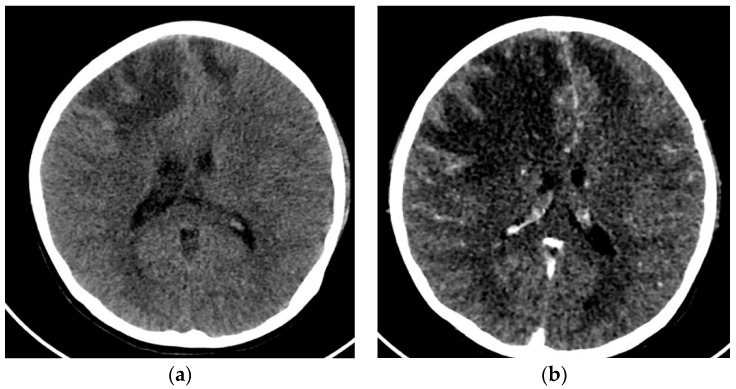
(**a**) Cranial CT with frontal venous infarction associated with a thrombosis of the superior longitudinal venous sinus. (**b**) Cranial CT with enlarged frontal ischemic areas and midline deviation.

**Figure 5 children-10-00852-f005:**
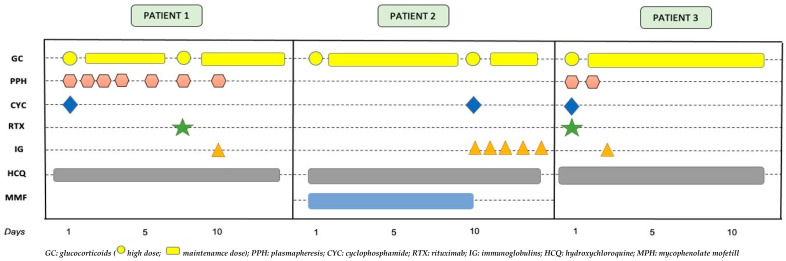
Temporal sequence of medical treatments received by the three patients.

## Data Availability

The data is contained within the article.

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
