# Peer review of "Severe Juvenile-Onset Systemic Lupus Erythematosus: A Case Series-Based Review and Update"

_children, 2023, doi:10.3390/children10050852_

Round 1
Reviewer 1 Report
Journal
Children
Title
Severe juvenile-onset systemic lupus erythematosus
Comments
This topic is interesting especially with the rheumatologist’s interest in pediatric rheumatology. Some points need to be addressed.
Abstract
but sometimes the onset…English editing and grammar revising are required in the whole manuscript.
The title should refer to the nature of the study and if it is a case series or case report and if it includes review of the literature.
Abstract should be more concise and at its conclusion should cope with the results.
Introduction
with a peak age of onset at 12.6 years [3]c…. What is meant by c after 3?
European evidence-based recommendations for management of jSLE…. the authors need to revise the references as many sentences without its reference.
The research gap should be concise and clear at the end of the introduction just before the aim of the study.
Case 1 ...the work up of infection should be written as the authors gave aggressive immunosuppression. In this situation ,it is mandatory to exclude infection.
Case 2 ...again infection was not excluded before given aggressive immunosuppression.
MRI was done and the authors did not refer to the presence of CNS manifestations in the case from the start.
Also, she receives acetylsalicylic acid…grammar revising is mandatory ...Also, it was strange to add antiplatelets in a case with suspected alveolar hemorrhage.
catastrophic antiphospholipid syndrome (CAPS)…it is not clear how the author suspect CAPS ...the authors should declare this as CAPS needs 3 organ thromboses in a short time to be diagnosed .
current EULARS clinical criteria for the diagnosis of this condition…. what is the value of S in EULARS
English editing and Grammer revision are mandatory in the whole manuscript
Reviewer 2 Report
This is an interesting Case report describing a couple of cases with severe onset of juvenile SLE. The information may be important to convey to clinicians. However, there are a number of typos and editing of English language in general is required.
I miss a short discussion on potential contribution of genetic factors, which may be more relevant in SLE with juvenile onset than in adult SLE.
, et al. S4D:5 Targeted next-generation sequencing suggests novel risk loci in juvenile onset systemic lupus erythematosus.There are a number of typos that should be corrected.
Round 2
Reviewer 1 Report
The manuscript improved
Grammar revision ws done
Author Response
Dear Reviewer,
Thank you very much for your feedback.
We have made a slight modification about the role of genetics based on a recommendation from the Academic Editor. In addition, we have modified a mistake where CAPS was mentioned instead of APS.
The manuscript was sent for an English revision.